# Rethinking the Future of Food Packaging: Biobased Edible Films for Powdered Food and Drinks

**DOI:** 10.3390/molecules24173136

**Published:** 2019-08-28

**Authors:** Roxana Puscaselu, Gheorghe Gutt, Sonia Amariei

**Affiliations:** Faculty of Food Engineering, Stefan cel Mare University of Suceava, University Street 13, 72229 Suceava, Romania

**Keywords:** inulin, biopolymers, edible packaging, food, environment

## Abstract

In today’s society, packaging is essential. Without this, the materials would be messy and ineffective. Despite the importance and key role of packaging, they are considered to be useless, as consumers see it as a waste of resources and an environmental threat. Biopolymer-based edible packaging is one of the most promising solutions to these problems. Thus, inulin, biopolymers such as agar and sodium alginate, and glycerol were used to develop a single use edible material for food packaging. These biofilms were obtained and tested for three months. For inulin-based films, the results highlight improvements not only in physical properties (homogeneity, well-defined margins, light sweet taste, good optical properties, high solubility capacity or, as in the case of some samples, complete solubilization), but also superior mechanical properties (samples with high inulin content into composition had high tensile strength and extremely high elongation values). Even after three months of developing, the values of mechanical properties indicate a strong material. The optimization establishes the composition necessary to obtain a strong and completely water-soluble material. This type of packaging represents a successful alternative for the future of food packaging: they are completely edible, biodegradable, compostable, obtained from renewable resources, and produce zero waste, at low cost.

## 1. Introduction

Nowadays, the total amount of plastic waste exceeds 200 million tons, with an annual increase of about 5%. It is extremely important to use alternative materials. Until recently, the petroleum-based materials used to make plastics were polyethylene terephthalate (PET), vinyl polychloride (PVC), polyethylene (PE), polypropylene (PP), polystyrene (PS) or polyamide (PA). They are heavily used due to low cost, and especially for mechanical performance: high tensile strength and elasticity, good barrier to O_2_, CO_2_, and aroma compounds, heat stability, etc. Plastic packaging usually contains residues from the food it contains and other biological substances, so their recycling is impractical and economically inconvenient; as a result, tons of plastic packaging are thrown into nature, thus increasing the waste problem year after year, [1]. In order to reduce these environmental problems, the solutions involve the use of both biodegradable and edible biopolymer packaging [2]. An extremely feasible alternative, both in terms of food safety and renewable resources, is the use of edible films or coatings. Edible films, with a low environmental impact, have been progressively improved to effectively protect various food products; manufacturers are trying to reduce the applications of plastic materials, mainly for food packaging, and to develop innovative edible films [3,4,5,6].

Conventional packages of pulverulent products are very harmful to the environment, especially because they are obtained from several layers (polypropylene, polyethylene, a metallic layer which is usually aluminum, as well as substances used for printing and gluing). For this reason, research has focused on developing completely edible packaging materials that can be fully solubilized in higher temperature water and can be eaten along with the packaged product [7]. Polysaccharides represent potential valuable raw-materials for next generation advanced and environmentally friendly plastics [7]. Agar and sodium alginate, as well as glycerol and chicory inulin have been used to obtain such edible packaging.

Due to its thermoplastic character, biodegradability, biocompatibility and moderate water resistance, agar has begun to be used as an alternative to conventional plastic materials. Agar has the low thermal and mechanical properties required for food packaging applications. To improve both the features and the cost, nanotechnology was applied to obtain biopolymer films [8]. Agar-based films are an ideal matrix for the incorporation of micronutrients, extracts or essential oils to obtaining biodegradable packaging materials, since they allow the release of antioxidant and antimicrobial compounds [9,10].

Sodium alginate can form films with specific properties: strength, gloss, without taste or smell, flexibility, water-solubility, low permeability to O_2_ and oils. In combination with glycerol, it has been used to coat fruits (cherries), contributing to the delay of degenerative processes, maintaining color, polyphenols and anthocyanins, and improving overall fruit quality after harvest [11]. Along with other hydrocolloids, they form coatings used in many sub-sectors of the food industry [12].

Inulin (C_6n_H_10n+2_O_5n+1_) is a widespread beta fructan and is found in over 3000 plant species (Figure 1).

Natural sources of inulin are chicory root, endives, dill, asparagus, leek, onion, garlic, banana or wheat [13]. Plant inulin has chains incorporating from two to 100 fructose units, whose length, composition and polydispersity depend on the plant species, the phase in its life cycle, the harvesting date and the extraction and post-extraction procedures [14]. The physicochemical and functional properties of inulin are linked to degree of polymerization and the presence of branches. The short-chain fraction, oligofructose, is much more soluble and sweeter than native and long-chain inulin, and can contribute to improve mouthfeel because its properties are closely related to those of other sugars. The long-chain is less soluble, more viscous and more thermostable than native inulin [14]. In addition to its ability to function as a fiber in the human diet (it cannot be hydrolyzed by digestive enzymes), it prevents digestive diseases, facilitates the absorption of calcium, magnesium and iron ions, regulates appetite and stimulates the immune system [15,16,17]. Due to the techno-functional attributes, inulin is used extensively in the industry to maintain constant humidity and for fiber intake (bread and bakery products [18,19,20]), to substitute sugar in the composition and to maintain the texture of crunchy particles (breakfast cereals [21,22]), as stabilizer, sugar substitute and for fiber intake (dairy products [23,24]), as a fat substitute and to prevent the reduction of mass cooking loss (meat products [25,26]), as a substitute for sugar, or fats, for texture improvement and resistance to heat treatment (frozen desserts, chocolate, fruit desserts [27,28,29]).

The substances used and the composition established for the development of edible films are safe for consumption. Thus, referring to the maximum permitted level of substances, both biopolymers—sodium alginate, agar—and glycerol can be used in quantum satis, according to EC Regulation 1129/2011 [30]. Sodium alginate, agar, and glycerol are well established as “generally recognized as safe” (GRAS) food additive in the United States [31].

Studies and research on toxicity tests conducted indicate the safety of the consumption of sodium alginate [32], agar [33], and glycerol [34].

The purpose of this research is to create an edible material that can be used to pack food that is sold in the form of powder and requires solubilization before consumption (like soluble coffee, cappuccino, powdered milk, spices, teas, or dehydrated fruits and vegetables). The evaluation of the biofilms obtained, as well as of the products packaged in these films, has strengthened the possibility of their use, as there have been no major changes in the initial parameters during the tested period. Nowadays, obtaining a completely biodegradable material (or edible, as is the case here) has become necessary due to the benefits to consumers and the environment [35].

## 2. Results and Discussions 

New materials used for packaging pulverulent products are particularly important because they produce “zero waste” and do not pollute the environment, being easily obtained from biodegradable and renewable resources. For packaging pulverulent products, conventional packaging is typically made of a plurality of layers of material (polyethylene, metallic foil, solvent-based flexographic ink, lamination and LDPE – low-density polyethylene), so sorting and recycling is often impossible (Figure 2). The newly obtained material can be stored in the supermarket in cardboard boxes.

Physical properties and applications for use: Observation of physical properties showed homogeneous, smooth films, without pores or cracks. The films have well-defined edges with very low adhesion to the silicone support used for drying, were odorless, but slightly sweet, transparent and glossy. Except for I2, all samples can be used as packing materials for low moisture products, such as instant drinks, dehydrated vegetables, films for the pharmaceutical industry (mainly due to its total solubility in a short time, 1 minute in liquid at room temperature). I5 sample could be used as self-adhesive foil for food, and I8 film may be used to obtain edible consumables. I12–I16 samples can be used as packaging/wrapping material for fresh fruit and vegetables, for the cosmetic industry (film impregnated with various substances), and medicine. I17 film is the most suitable to be used as packaging material for powdered food (in terms of physical properties), and I18–I20 samples could be used for packing ready-to-eat products, fast food, or ready-to-eat meats.

The values obtained after testing the transmittance evaluation sample were high (Table 1). Low content alginate films, I10 and I13 (1.00 g), showed significantly lower transmittance values than the others. The lowest transmittance was observed at sample I20, without alginate in composition (56.75%), and the highest at I4, with the highest content of alginate (3.00 g). We can deduce that the transmittance value increases directly proportional to the amount of alginate used to obtain the films. Therefore, the use of these films for packing foods with a high fat content is not recommended because it allows oxidative degradation in the presence of light. High values are not an impediment for powdered products, which, in addition to packaging in this single use material, are stored in cardboard boxes. For products with high fat content that are individually marketed, the films require improvements to enhance the opacity of the material; this can be achieved by adding various substances that can be successfully incorporated into the membrane matrix (carob powder, saffron, caramel, anthocyanins, carotenoids, or blackberries [36]).

Sample I7, considered optimal as packing material (as can be seen from the results), has medium tensile strength, unlike the other tested films (except for I5). The roughness of samples with sodium alginate addition exceeds 200.00 nm, unlike the roughness of agar-based films that have lower values. Increasing inulin and glycerol content leads to increased roughness (I18–I20).

The microstructure of the membranes indicates smooth, homogeneous films, without obvious pores or cracks, with a smooth and regular surface (Figure 3). These features indicate the possibility of using the material in an industry where the appearance matters in choosing the product by the consumer. The absence of pores and cracks highlights the quality of the film, but also its stability to the humidity or light in the environment.

There were no changes in physical characteristics; general appearance, color, taste or odor properties (Table 2).

The Pearson correlation (Table 2) of the film parameters with inulin addition does not indicate major changes throughout the test period (three months). According to Pearson correlation (Table 2), strong positive correlations can be observed between thickness, retraction ratio, but also tensile strength.

Negative correlations are identified between thickness and solubility. Samples with inulin addition are more resistant in the case of sample I7, the breaking strength could not be determined because it proved to be stronger than the established parameters for determination. The elongation at break showed extremely high values: initially exceeding the evaluation threshold, it was over 380% after one month. The results indicate that the addition of sodium alginate provides tear resistance and very good elongation at break. The I20 sample, with high inulin and glycerol content, exhibits the lowest tensile strength (although the elongation value was very high), a normal aspect when taking into account the absence of sodium alginate and the low amount of agar.

The evaluation of the films obtained, as well as of the products packed in this material, has strengthened the possibility of their use, as there were no major changes in the initial parameters during the three months of testing.

In order to create a perfect film regarding mechanical properties, the Design Expert 11 program was used for statistical interpretation and generated solution 1 of 43 for 1 desirability: tensile strength 1.13 MPa, elongation 380.13%, and roughness 29.37 nm (Figure 4 and Figure 5).

In order to obtain a film with high strength and elasticity, but reduced roughness, the optimization software generated the following composition: 0.4971 g, alginate = 1.018 g, glycerol = 1.797 g and inulin = 2.352 g (Figure 5).

Maintenance of physical characteristics over time demonstrates that the material can be successfully used in industry: During the three months tested time there were small variations in thickness and implicitly in the retraction ratio (Figure 6 and Figure 7).

To test the use of this material for the packaging of pulverulent products, in particular instant drinks (soluble coffee, cappuccino, cocoa, powdered milk) and its consumption with the product, the solubility of the samples was evaluated.

Samples I1–I7 could not be tested for the swelling ratio index (Figure 8) due to their complete dissolution, even after 1 minute of water immersion. The other samples immediately dissolved in hot water (over 80 °C), which promotes them for use as instant beverage packaging. Samples I14–I17, without the addition of sodium alginate but approximately equal agar mass, exhibited the lowest values of the swelling ratio index (showing the same behavior throughout the test period). This proves the high solubilization capacity of sodium alginate once again. By default, samples I8 and I11, with high agar content in their composition, were those which absorbed the largest amount of liquid.

All tested samples maintained their tendency during the three months of testing. This behavior qualifies the material for food products packaging. Similarly, other foods that require solubilization before consumption may be preserved: dehydrated vegetables and fruits, instant soups, vegetable concentrates, etc.

From the point of view of the incidence of microorganisms, the tested samples proved to be safe for consumption. No microorganisms from the tested ones (coliforms, enterobacterias, *E. coli, Staphylococcus aureus*, yeasts and molds) grew on culture media, with determinations carried out throughout the test period. If we also take into account the reduced water activity index (below 0.4), we can emphasize the safety of the films obtained.

## 3. Materials and Methods

### 3.1. Materials

Agar, sodium alginate, inulin from chicory root, glycerol and water were used to obtain the films. Except for agar (which was made available by ‘B & V The agar company’, Parma, Italy), the other products were purchased from Sigma Aldrich Company (Romania Order Center, Bucharest, Romania).

### 3.2. Methods

#### 3.2.1. Film Development

The films were made by cast method, using different proportions of biopolymers, plasticizers and inulin, according to Table 3. For each combination made the samples were obtained in triplicate, For the development of a 150.00 mL film forming solution, the composition used in the trials was as follows: 0–3.00 g agar, 0–3.00 g sodium alginate, 1.00–2.00 g glycerol, and 1.00–3.00 g inulin. The solution thus obtained was maintained for 30 min at 90 °C with continuous stirring; it was poured, leveled and left to dry on a silicone surface at a temperature of 23 ± 2 °C for about 48 h. For a correct determination of their behavior over time, the samples were evaluated for three months.

#### 3.2.2. Determination of Physical and Optical Properties

In order to evaluate the physical characteristics of each film, determinations such as adhesion to the drying surface, thickness, retraction ratio, and color were made. The thickness was measured with a precision of 0.001 mm electronic digital micrometer (Mitutoyo, Kawasaki, Japan) in at least seven random locations, and the average thickness was noted and taken into consideration for film evaluation.

Retraction ratio (R) is an important parameter for the reproducibility of determinations at industrial scale. The thickness of the final material can be predicted by knowing the thickness of the film-forming solution and the value of the retraction ratio.

The following formula was used for calculating the retraction ratio [37]:(1)$$R,\;\left(\%\right)=\frac{\mathrm{thickness}\;\mathrm{of}\;\mathrm{the}\;\mathrm{film}-\mathrm{forming}\;\mathrm{solution}\;-\mathrm{dry}\;\mathrm{film}\;\mathrm{thickness}}{\mathrm{thickness}\;\mathrm{of}\;\mathrm{the}\;\mathrm{film}-\mathrm{forming}\;\mathrm{solution}\;}\times 100$$

The results indicated represent the sum of five determinations, in different areas on the surface of the film tested.

The color was evaluated by CieLAB system using the Chroma Meter CR400 colorimeter (Konica Minolta, Tokyo, Japan), and the result was noted after at least five readings taken in different film regions. In addition, film transmittance was also evaluated at the wavelength of 660 nm using the Ocean Optics HR 4000 CG-UV-NIR spectrometer (Ocean Optics, 830 Douglas Ave., USA). Transmittance is an important feature of the material intended for the packaging of products that may be affected by light. The data obtained is of interest because, in the supermarket, the products exposed to the shelf are exposed to artificial light which can affect quality if it manages to cross the barrier of the packaging material.

#### 3.2.3. Determination of Mechanical Properties

For the mechanical performance evaluation, the samples were tested for tensile strength and elongation at break with an ESM 301 - Mark 10 texturometer (Stefan cel Mare University, Suceava, Romania), using the gripping attachments for thin films and sheets (Addex Design, Sibiu, Romania); the machine load was 5kN and the speed was set at 10 mm/min. For determination purposes, three samples of film strips were cut with dimensions of 100 mm × 10 mm, according to STAS ASTM D882 (Standard Test Method for Tensile Properties of Thin Plastic Sheeting) [38]. Tests were performed at ambient temperature of 28.4 °C. Tensile strength was calculated according to (2), where *F* represent de maximum applied force and *S* is the cross-section:(2)$$\mathrm{TS},\;\left(\mathrm{MPa}\right)=\frac{F}{S}$$

Elongation at break (3) is the ratio between increased length (*Δl*) and initial length (*l*) and represents the ability of samples to withstand changes without cracking.
(3)$$E,\;\left(\%\right)=\frac{\Delta l}{l}\times 100$$

Roughness and films’ microstructures were evaluated using the Mahr CWM100 microscope (Mahr, Gottingen, Germany), the results being noted after observing at least five different areas. In order to analyze the microscopic structure and reflectivity of film surface Mountain Map^®^ software (Version 7, Digital Surf, Lavoisier, France)—surface imaging, analysis, and metrology software—was used.

All tests were performed in triplicate.

#### 3.2.4. Assessment of Solubility

Since the material obtained is desired to be used for the packaging of powdered products with complete dissolution in hot water, solubility is an important parameter. In order to evaluate it, a series of determinations were carried out to characterize the material from this point of view: moisture content, water solubility, swelling ratio and water activity index were measured. For moisture determination, film samples (3 × 3 cm) were weighed and held for 24 h at 110 °C, re-weighted and the results were noted in the moisture calculation (4) formula [39]:(4)$$\mathrm{MC},\;\left(\%\right)=\frac{W_{0}-W_{1}}{W_{0}}\times 100$$where *w*_0_ represents the weight of initial sample, and *w*_1_ the weight of dry sample.

Solubility in water (*WS*) implied the use of the same measure of sample; thus, 3 × 3 cm pieces were cut, weighed (*w*_0_), immersed in 22 °C water for 8 h, dried in a hot air oven (Memmert, Schwabach, Germany) for 24 h at 110 °C, and then re-weighted (*w*_1_) [39]. Water solubility was calculated according to following Formula (5):(5)$$WS,\;\left(\%\right)=\frac{W_{0}-W_{1}}{W_{0}}\times 100$$

Swelling ratio: For this determination, the films were immersed in water at 22 °C within a time interval of 1–20 min. Samples were weighed before and after immersion, and the value of the retraction ratio was determined using the Formula (6):(6)$$SR,\;\left(\%\right)=\frac{W_{t}-W_{0}}{W_{0}}\times 100$$where *SR* represent the swelling ratio, *W_t_* is the film mass at t moment, [g] and *W*_0_ represent the initial mass in grams [40].

The water activity index (aw) was evaluated using the AquaLab equipment (ICT International Amirdale, NSW 2350, Palestine) and determinations were made at 22.8 ± 2 °C. The value noted represents the sum of five determinations.

To achieve the determinations and obtain the correct results, the film samples were kept under controlled temperature and humidity conditions, 42% RH and 20 ± 2 degrees respectively.

All determinations were performed in triplicate.

#### 3.2.5. Evaluation of Microbiological Characteristics

When a food or other product is ingested, it must be microbiologically safety. Thus, both the obtained films and the biopolymers, glycerol, and inulin used were tested for the identification of coliforms, enterobacteria, *Escherichia coli*, *Staphylococcus aureus*, and yeasts and molds. For this purpose, Compact Dry TC/CF/ETB/EC/XSA/YM dehydrated specific culture media were used (NISSUI Pharmaceutical, Tokyo, Japan) and thermostat conditions for each microorganism were respected. The principle of the method involved the solubilization of 1 g of the test film in 9 mL of physiological saline; 1 mL of the newly obtained solution was poured onto the plates with specific culture medium and kept for thermostation at 37 °C for 48 h (in the case of total count, enterobacteria, *E. Coli*, and *S. Aureus*), and 72 h for yeasts and molds.

All microbiological tests were performed in triplicate.

### 3.3. Statistical Analyses

Statistical analyses were performed by IBM SPSS vers. 25.0. The mean differences between groups were estimated by Tukey test and compared using the one way analysis of variance (ANOVA). Differences were considered significant at *p-value* < 0.05. To optimize the composition, the software Design Expert^®^ 11 (trial version) was used.

## 4. Conclusions

Edible films can be used in the food industry and other adjacent areas for packaging products requiring solubilization or rehydration prior to use.

Keeping qualities during the test period indicates the safety of product packaging in this type of material. Due to the advantages in use (fiber, carbohydrate and fat replacement), inulin was used to produce packaging for powdered products. The addition of inulin improved the appearance, mechanical properties (elongation) and facilitated solubilization. In the case of sample I7, the tensile strength could not be determined because it proved to be higher than the parameters set for determination; the elongation presented extremely high values, initially above the possibility of evaluation, and after one month with a value of over 380%). This material can replace foil for freshness due to the self-adhesive nature of the gloss, transparency, homogeneity, well-defined margins, and medium solubility. The sample with maximum inulin content in composition (I17)—3.00 g and no agar added—has optimal characteristics for use as packaging material for pulverulent products: it was high gloss and flexible, thin, without pores or cracks, with regular margins, homogeneous, odorless, with a sweet taste, high solubility, medium roughness (158.25 nm) and very good elasticity (149.40%). Samples with high agar content in the composition, medium inulin and without sodium alginate can be used to pack sliced fruits and vegetables or ready-to-eat sausages due to their low solubility; they were flexible, soft, shiny, without pores or cracks, with low transmittance values (56.75% in I20), medium to high roughness (due to the large amount of inulin), good tensile strength, and high elasticity.

Due to increased solubility and mechanical strength, inulin is a valuable ingredient for the production of biopolymers-based edible materials intended for food packaging.

Depending on the need and use, the industry can control the amount of added inulin so that the desirable product results.

## Aknowledgments

The authors are thankful to the EDR Ingredients Romania and B&V The Agar Company Italia for their support to the development of this study and for providing the materials necessary for successfully conclude this research.

## Figures and Tables

**Figure 1 molecules-24-03136-f001:**
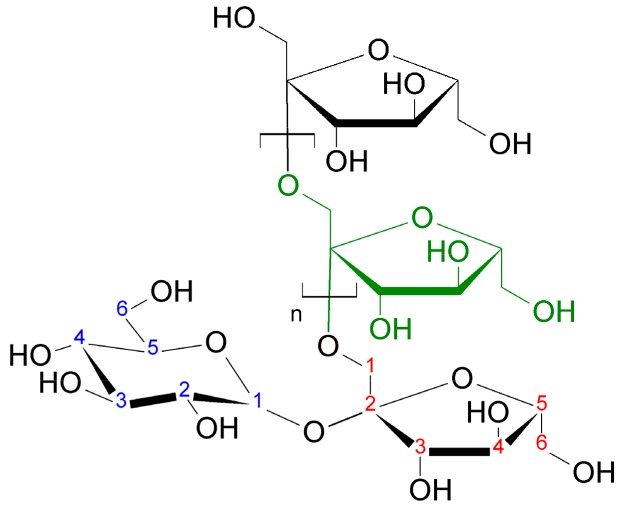
The general structural formula of inulin.

**Figure 2 molecules-24-03136-f002:**
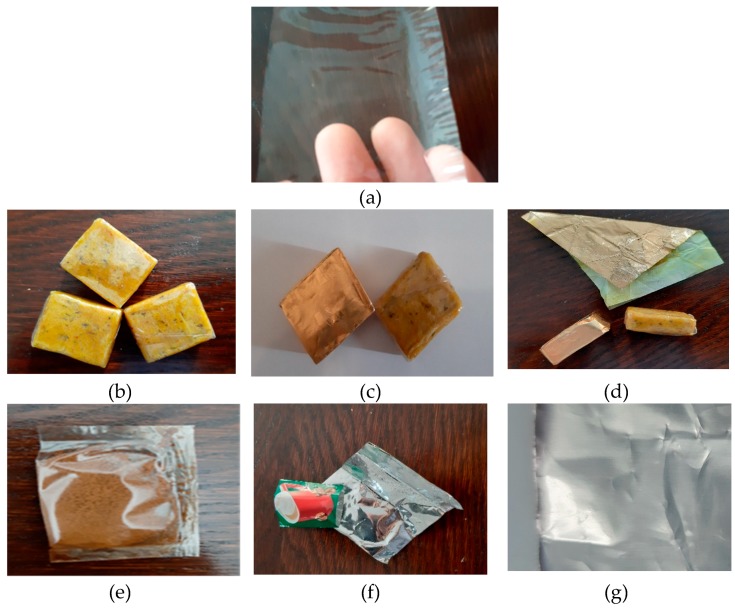
Applications of edible films. Replacement of conventional multilayer packaging with edible materials. (**a**) Film enriched with inulin (I17), (**b**) dehydrated vegetables packed in edible foil, (**c**,**d**) differences between conventional packaging and new packaging, (**e**) soluble coffee packed in edible foil, (**f**,**g**) metal foil used for soluble coffee packaging.

**Figure 3 molecules-24-03136-f003:**
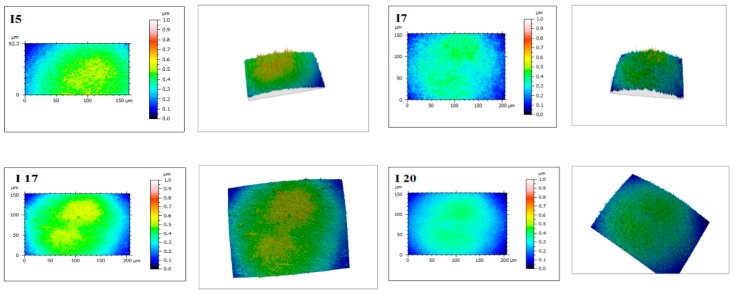
Microscopic structures and reflectivity of films’ surfaces.

**Figure 4 molecules-24-03136-f004:**
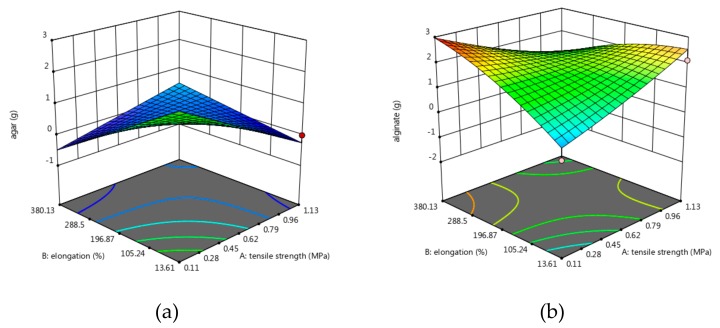
Optimization for a material with high strength and elasticity, but minimal roughness. (**a**) agar content, (**b**) alginate content, (**c**) glycerol content, (**d**) inulin content.

**Figure 5 molecules-24-03136-f005:**
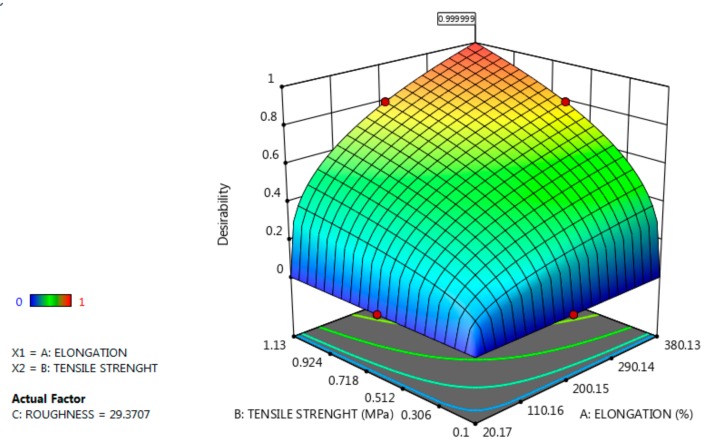
Values of mechanical properties in order to obtain a resistant and elastic film with minimal roughness.

**Figure 6 molecules-24-03136-f006:**
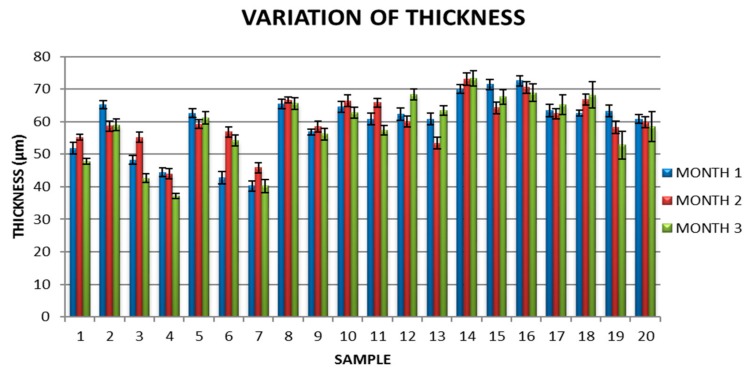
Variation of film thickness during tested time.

**Figure 7 molecules-24-03136-f007:**
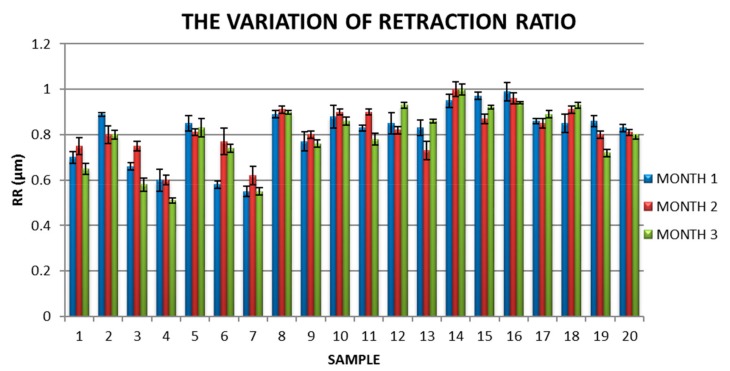
Variation of the retraction ratio during the three months of testing.

**Figure 8 molecules-24-03136-f008:**
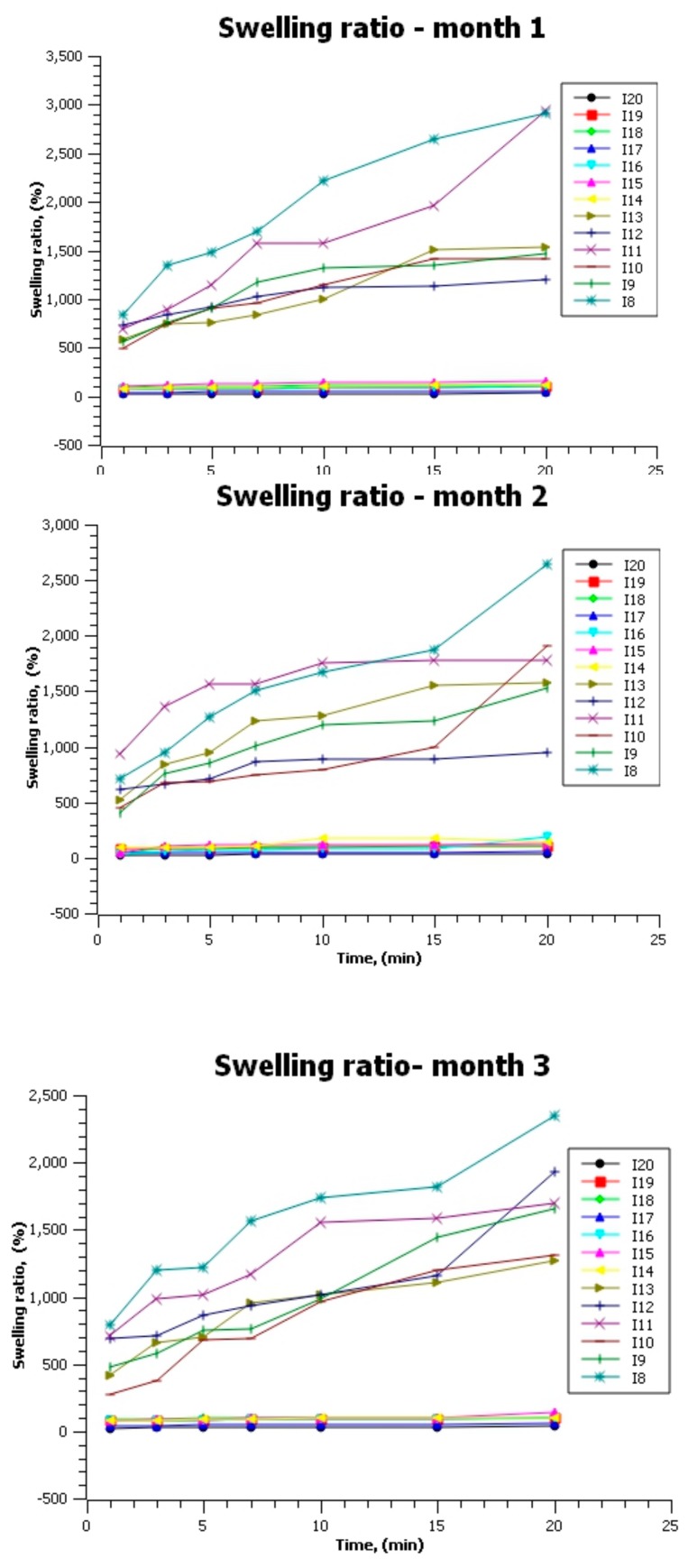
The graphical representation of the swelling ratio index over the three months of testing position.

**Table 1 molecules-24-03136-t001:** Tested films characteristics.

Sample	Transmittance, T, (%)	Roughness, R_z_, (nm)	Water Activity Index, (a_w_)
**I1**	82.44 ^a,d,g^ ± 2.1	144.00 ^a,c^ ± 0.05	0.376 ^a,b,c^ ± 0.07
**I2**	84.42 ^a,d,h^ ± 1.01	276.40 ^a,b^ ± 0.07	0.387 ^a^ ± 0.05
**I3**	89.68 ^b^ ± 0.7	211.20 ^c,g^ ± 0.02	0.371 ^a^^,b,c^ ± 0.01
**I4**	99.43 ^c^ ± 0.41	371.66 ^d^ ± 0.33	0.374 ^a,b,c.^ ± 0.97
**I5**	85.09 ^a,b^ ± 1.7	29.37 ^e^ ± 0.57	0.368 ^a,b,c^ ± 0.47
**I6**	97.89 ^c^ ± 0.62	108.36 ^c,e,f^ ± 0.97	0.387 ^a,b^ ± 1.02
**I7**	80.21 ^d,g^ ± 2.2	90.36 ^b,g^ ± 1.04	0.368 ^a,b,c^ ± 0.98
**I8**	80.44 ^a,d,g^ ± 2.11	156.25 ^b,h^ ± 0.61	0.375 ^a,b,c^ ± 0.57
**I9**	89.48 ^b^ ± 0.73	222.33 ^i^ ± 0.84	0.375 ^a,b,c^ ± 0.78
**I10**	66.30 ^e^ ± 2.63	207.47 ^i^ ± 0.77	0.368 ^a,b,c^ ± 0.25
**I11**	95.26 ^c^ ± 0.52	224.25 ^i^ ± 0.07	0.363 ^a,b,c^ ± 0.77
**I12**	73.31 ^f^ ± 2.4	213.50 ^b,i^ ± 0.33	0.361 ^b,c^ ± 0.68
**I13**	65.36 ^e^ ± 2.19	195.50 ^b,i^ ± 0.04	0.356 ^c^ ± 0.62
**I14**	68.52 ^e,f^ ± 0.97	202.25 ^b,i^ ± 0.93	0.369 ^a,b,c^ ± 0.05
**I15**	82.08 ^a,d,g^ ± 3.21	158.00 ^c^ ± 0.61	0.369 ^a,b,c^ ± 0.21
**I16**	95.89 ^c^ ± 1.34	171.60 ^g,h^ ± 0.05	0.364 ^a,b,c^ ± 0.41
**I17**	84.61 ^a,d,h^ ± 4.77	158.25 ^g^ ± 0.77	0.361 ^b,c,d^ ± 0.33
**I18**	78.36 ^g^ ± 1.63	228.75 ^i^ ± 0.33	0.357 ^c,^ ± 0.18
**I19**	88.89 ^b,h^ ± 0.66	299.00 ^d^ ± 0.57	0.367 ^b,c^ ± 0.57
**I20**	16.76 ^i^ ± 1.88	224.00 ^i^ ± 0.02	0.356 ^c^ ± 0.48

**Table 2 molecules-24-03136-t002:** Pearson correlation for characteristics of tested films.

**MONTH 1**		**T**	**R**	**TS**	**E**	**MC**	**WS**	**L***	**a***	**b***
	**T**	1	0.999 *	0.490	0.675	−0.353	−0.644	−0.224	−0.145	0.003
	**R**		1	−0.780 **	0.597	0.351	−0.646	−0.221	−0.147	0.002
	**TS**			1	0.696 **	−0.434	−0.410	−0.547	0.347	0.596
	**E**				1	−0.192	−0.120	0.159	−0.192	−0.267
	**MC**					1	0.567	0.839 *	−0.611	−0.748 **
	**WS**						1	0.630	−0.308	−0.464
	**L***							1	−0.839 *	−0.949 *
	**a***								1	0.844 *
	**b***									1
**MONTH 2**		**T**	**R**	**TS**	**E**	**MC**	**WS**	**L***	**a***	**b***
	**T**	1	0.999 *	−0.524	−0.472	−0.590	−0.609	−0.381	0.366	0.138
	**R**		1	−0.520	−0.473	−0.570	−0.615	−0.388	0.362	0.140
	**TS**			1	0.663 **	−0.07	0.254	−0.172	0.166	0.306
	**E**				1	0.475	0.355	0.477	−0.389	−0.501
	**MC**					1	0.106	0.278	0.243	−0.174
	**WS**						1	0.681	−0.598	−0.352
	**L***							1	−0.523	−0.794 *
	**a***								1	0.474
	**b***									1
**MONTH 3**		**T**	**R**	**TS**	**E**	**MC**	**WS**	**L***	**a***	**b***
	**T**	1	0.999 *	0.808 *	−0.408	−0.610	−0.655	−0.224	0.197	0.090
	**R**		1	0.812 *	−0.409	−0.020	−0.655	0.207	0.207	0.096
	**TS**			1	−0.248	0.143	−0.124	−0.114	0.207	0.097
	**E**				1	0.176	0.491	0.585	−0.457	−0.571
	**MC**					1	0.221	0.267	−0.052	−0.310
	**WS**						1	0.544	−0.433	−0.531
	**L***							1	−0.762 *	−0.895 **
	**a***								1	0.574
	**b***									1

T-thickness, (µm), R-retraction ratio (%), TS-tensile strength, (MPa), E-elongation, (%), MC-moisture content, (%), S-solubility, (%), L*, a*,b*-CieLAB color parameters; * Correlation is significant at the 0.05 level; ** Correlation is significant at the 0.01 level.

**Table 3 molecules-24-03136-t003:** The composition of edible films.

Sample	m_ALGINATE_, (g)	m_INULIN_, (g)	m_AGAR_, (g)	m_GLYCEROL_, (g)	V_WATER_ (mL)
I_1_	2.50	2.00	0.00	1.00	150.00
I_2_	2.00	2.50	0.00	1.00	
I_3_	2.00	2.00	0.00	1.50	
I_4_	3.00	1.00	0.00	1.50	
I_5_	1.00	3.00	0.00	1.50	
I_6_	2.50	1.00	0.00	2.00	
I_7_	1.00	2.50	0.00	2.00	
I_8_	2.50	1.00	1.00	1.00	
I_9_	1.00	2.50	1.00	1.00	
I_10_	1.00	1.00	2.50	1.00	
I_11_	2.00	1.00	1.00	1.50	
I_12_	1.00	2.00	1.00	1.50	
I_13_	1.00	1.00	2.00	1.50	
I_14_	0.00	2.00	2.50	1.00	
I_15_	0.00	2.50	2.00	1.00	
I_16_	0.00	2.00	2.00	1.50	
I_17_	0.00	3.00	1.00	1.50	
I_18_	0.00	1.00	3.00	1.50	
I_19_	0.00	1.00	2.50	2.00	
I_20_	0.00	2.50	1.00	2.00	

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
