# Peer review of "Rethinking the Future of Food Packaging: Biobased Edible Films for Powdered Food and Drinks"

_molecules, 2019, doi:10.3390/molecules24173136_

Round 1

Reviewer 1 Report

The manuscript under consideration deals with biopolymers-based edible materials intended for food packaging. It is interesting contribution and worth to be published after minor revision where authors should address the toxicity studies of the materials described as well as the potential interactions with the food content.

Author Response

Dear Editor,

Thank You for the review.

At your suggestion, we have added information regarding the acceptability of the substances used and toxicity studies.

Respectfully, 

Roxana Puscaselu

Reviewer 2 Report

Detailed recommendation:

Page 1, line 32 – 2 in subscript

Page 1, lines 1-40 – please add more references

Page 2, line 56 – 2 in subscript

Page 2, lines 62-72 – information about chemical structure and properties of inulin are necessary

Table 1 – the information about statistically differences in results are missing. It should be improved.

Page 11, lines 212, 234, 243, 251, 256 – in how many replicates were the analyzes performed?

Page 2, line 273 – bacterial names should be in italics

Page 13, line 312 – add Author Contributions

Author Response

Dear Reviewer,

Thank You for the review and suggestions.

According to your recommendations, we have completed detailes regarding the chemical structure and properties of inulin. Also, where changes were needed in the text (bacterial names in italics, subscript letters), we made it.

Regarding the number of replicates, it is written in the text preceding the formula: to evaluate the color parameters and to establish the water activity index, it were made five readings in different areas of film's surface; mechanical tests were made in triplicate, and solubility assessments were performed in triplicate. 

Respectfully,

Roxana Puscaselu